# DECENTRALIZED POLICY OPTIMIZATION

## ABSTRACT

The study of decentralized learning or independent learning in cooperative multi-agent reinforcement learning has a history of decades. Recently empirical studies show that independent PPO (IPPO) can obtain good performance, close to or even better than the methods of centralized training with decentralized execution, in several benchmarks. However, decentralized actor-critic with convergence guarantee is still open. In this paper, we propose *decentralized policy optimization* (DPO), a decentralized actor-critic algorithm with monotonic improvement and convergence guarantee. We derive a novel decentralized surrogate for policy optimization such that the monotonic improvement of joint policy can be guaranteed by each agent *independently* optimizing the surrogate. In practice, this decentralized surrogate can be realized by two adaptive coefficients for policy optimization at each agent. Empirically, we compare DPO with IPPO in a variety of cooperative multi-agent tasks, covering discrete and continuous action spaces, and fully and partially observable environments. The results show DPO outperforms IPPO in most tasks, which can be the evidence for our theoretical results.

## 1 INTRODUCTION

In cooperative multi-agent reinforcement learning (MARL), centralized training with decentralized execution (CTDE) has been the primary framework (Lowe et al., 2017; Foerster et al., 2018; Sunehag et al., 2018; Rashid et al., 2018; Wang et al., 2021a; Zhang et al., 2021; Yu et al., 2021). Such a framework can settle the non-stationarity problem with the centralized value function, which takes as input the global information and is beneficial to the training process. Conversely, decentralized learning has been paid much less attention. The main reason may be that there are few theoretical guarantees for decentralized learning and the interpretability is insufficient even if the simplest form of decentralized learning, *i.e.*, independent learning, can obtain good empirical performance in several benchmarks (Papoudakis et al., 2021). However, decentralized learning itself still deserves attention as there are still many settings in which the global information cannot be accessed by each agent, and also for better robustness and scalability (Zhang et al., 2019). Moreover, the idea of decentralized learning is direct, comprehensible, and easy to realize in practice.

Independent Q-learning (IQL) (Tampuu et al., 2015) and independent PPO (IPPO) (de Witt et al., 2020) are the straightforward decentralized learning methods for cooperative MARL, where each agent learns the policy by DQN (Mnih et al., 2015) and PPO (Schulman et al., 2017) respectively. Empirical studies (de Witt et al., 2020; Yu et al., 2021; Papoudakis et al., 2021) demonstrate that these two methods can obtain good performance, close CTDE methods. Especially, IPPO can outperform several CTDE methods in a few benchmarks, including MPE (Lowe et al., 2017) and SMAC (Samvelyan et al., 2019), which shows great promise for decentralized learning. Unfortunately, to the best of our knowledge, there is still no theoretical guarantee or rigorous explanation for IPPO, though there has been some study (Sun et al., 2022).

In this paper, we make a further step and propose *decentralized policy optimization* (**DPO**), a decentralized actor-critic method with monotonic improvement and convergence guarantee for cooperative multi-agent reinforcement learning. Similar to IPPO, DPO is actually *independent learning*, because in DPO each agent optimizes its own objective individually and independently. However, unlike IPPO, such an independent policy optimization of DPO can guarantee the monotonic improvement of the joint policy.

From the essence of fully decentralized learning, we first analyze Q-function in the decentralized setting and further show that the optimization objective of IPPO may not induce the joint policy

improvement. Then, starting from the surrogate of TRPO (Schulman et al., 2015) and together considering the characteristics of fully decentralized learning, we introduce a *novel* lower bound of joint policy improvement as the surrogate for decentralized policy optimization. This surrogate can be naturally decomposed for each agent, which means each agent can optimize its individual objective to make sure that the joint policy improves monotonically. In practice, this decentralized surrogate can be realized by two adaptive coefficients for policy optimization at each agent. The idea of DPO is simple yet effective, and suitable for fully decentralized learning.

Empirically, we compare DPO and IPPO in a variety of cooperative multi-agent tasks including a cooperative stochastic game, MPE (Lowe et al., 2017), multi-agent MuJoCo (Peng et al., 2021), and SMAC (Samvelyan et al., 2019), covering discrete and continuous action spaces, and fully and partially observable environments. The empirical results show that DPO performs better than IPPO in most tasks, which can be evidence for our theoretical results.

## 2 RELATED WORK

**CTDE.** In cooperative MARL, centralized training with decentralized execution (CTDE) is the most popular framework (Lowe et al., 2017; Iqbal & Sha, 2019; Foerster et al., 2018; Sunehag et al., 2018; Rashid et al., 2018; Wang et al., 2021a; Zhang et al., 2021; Peng et al., 2021). CTDE algorithms can handle the non-stationarity problem in the multi-agent environment by the centralized value function. One line of research in CTDE is value decomposition (Sunehag et al., 2018; Rashid et al., 2018; Son et al., 2019; Yang et al., 2020; Wang et al., 2021a), where a joint Q-function is learned and factorized into local Q-functions by the relationship between optimal joint action and optimal local actions. Another line of research in CTDE is multi-agent actor-critic (Foerster et al., 2018; Iqbal & Sha, 2019; Wang et al., 2021b; Zhang et al., 2021; Su & Lu, 2022), where the centralized value function is learned to provide policy gradients for agents to learn stochastic policies. More recently, policy optimization has attracted much attention for cooperative MARL. PPO (Schulman et al., 2017) and TRPO (Schulman et al., 2015) have been extended to multi-agent settings by MAPPO (Yu et al., 2021), CoPPO (Wu et al., 2021), and HAPPO (Kuba et al., 2021) respectively via learning a centralized state value function. However, these methods are CTDE and thus not appropriate for decentralized learning.

**Fully decentralized learning.** Independent learning (OroojlooyJadid & Hajinezhad, 2019) is the most straightforward approach for fully decentralized learning and has actually been studied in cooperative MARL since decades ago. The representatives are independent Q-learning (IQL) (Tan, 1993; Tampuu et al., 2015) and independent actor-critic (IAC) as Foerster et al. (2018) empirically studied. These methods make agents directly execute the single-agent Q-learning or actor-critic algorithm individually. The drawback of such independent learning methods is obvious. As other agents are also learning, each agent interacts with a non-stationary environment, which violates the stationary condition of MDP. Thus, these methods are not with any convergence guarantee theoretically, though IQL could obtain good performance in several benchmarks (Papoudakis et al., 2021). More recently, decentralized learning has also been specifically studied with communication (Zhang et al., 2018; Li et al., 2020) or parameter sharing (Terry et al., 2020). However, in this paper, we consider fully decentralized learning as each agent independently learning its policy while being not allowed to communicate or share parameters as in Tampuu et al. (2015); de Witt et al. (2020). We will propose an algorithm with convergence guarantees in such a fully decentralized learning setting.

**IPPO.** TRPO (Schulman et al., 2015) is an important single-agent actor-critic algorithm that limits the policy update in a trust region and has a monotonic improvement guarantee by optimizing a surrogate objective. PPO (Schulman et al., 2017) is a practical but effective algorithm derived from TRPO, which replaces the trust region constraint with a simpler clip trick. IPPO (de Witt et al., 2020) is a recent cooperative MARL algorithm in which each agent just learns with independent PPO. Though IPPO is still with no convergence guarantee, it obtains surprisingly good performance in SMAC (Samvelyan et al., 2019). IPPO is further empirically studied by Yu et al. (2021); Papoudakis et al. (2021). Their results show IPPO can outperform a few CTDE methods in several benchmark tasks. These studies demonstrate the potential of policy optimization in fully decentralized learning, which we will focus on in this paper. *Although there are some value-based algorithms for fully decentralized learning (Tan, 1993; Matignon et al., 2007; Palmer et al., 2017), most of them are heuristic and follow the principle different from policy-based algorithms, so we will not focus on these algorithms.*

## 3 METHOD

From the perspective of policy optimization, in fully decentralized learning, we need to find an objective for each agent such that the joint policy improvement can be guaranteed by each agent independently and individually optimizing its own objective. Thus, we propose a novel lower bound of the joint policy improvement to enable *decentralized policy optimization* (DPO). In the following, we first discuss some preliminaries; then we analyze the critic of agent in fully decentralized learning; next we derive the lower bound and the proof for convergence; finally we introduce the practical algorithm of DPO.

### 3.1 PRELIMINARIES

**Dec-POMDP.** Decentralized partially observable Markov decision process is a general model for cooperative MARL. A Dec-POMDP is a tuple $\mathcal{G} = \{S, A, P, Y, O, I, N, r, \gamma\}$. $S$ is the state space, $N$ is the number of agents, $\gamma \in [0, 1)$ is the discount factor, and $I = \{1, 2 \cdots N\}$ is the set of all agents. $A = A_1 \times A_2 \times \cdots \times A_N$ represents the joint action space, where $A_i$ is the individual action space for agent $i$. $P(s'|s, \boldsymbol{a}) : S \times A \times S \rightarrow [0, 1]$ is the transition function, and $r(s, \boldsymbol{a}) : S \times A \rightarrow \mathbb{R}$ is the reward function of state $s \in S$ and joint action $\boldsymbol{a} \in A$. $Y$ is the observation space, and $O(s, i) : S \times I \rightarrow Y$ is a mapping from state to observation for each agent $i$. The objective of Dec-POMDP is to maximize $J(\boldsymbol{\pi}) = \mathbb{E}_{\boldsymbol{\pi}}\left[\sum_{t=0} \gamma^t r(s_t, \boldsymbol{a}_t)\right]$, thus we need to find the optimal joint policy $\boldsymbol{\pi}^* = \arg\max_{\boldsymbol{\pi}} J(\boldsymbol{\pi})$. To settle the partial observable problem, history $\tau_i \in \mathcal{T}_i = (Y \times A_i)^*$ is often used to replace observation $o_i \in Y$. In fully decentralized learning, each agent $i$ independently learns an individual policy $\pi^i(a_i|\tau_i)$ and their joint policy $\boldsymbol{\pi}$ can be represented as the product of each $\pi^i$. Though each agent learns individual policy as $\pi^i(a_i|\tau_i)$ in practice, in our analysis, we will assume that each agent could receive the state $s$, because the analysis in partially observable environments is much more difficult and the problem may be undecidable in Dec-POMDP (Madani et al., 1999). Moreover, the V-function and Q-function of the joint policy $\boldsymbol{\pi}$ are as follows,

$$V^{\boldsymbol{\pi}}(s) = \mathbb{E}_{\boldsymbol{a} \sim \boldsymbol{\pi}}\left[Q^{\boldsymbol{\pi}}(s, \boldsymbol{a})\right] \tag{1}$$

$$Q^{\boldsymbol{\pi}}(s, \boldsymbol{a}) = r(s, \boldsymbol{a}) + \gamma \mathbb{E}_{s' \sim P(\cdot|s, \boldsymbol{a})}\left[V^{\boldsymbol{\pi}}(s')\right]. \tag{2}$$

**Joint TRPO Objective.** In Dec-POMDP, we can still obtain a TRPO objective for the joint policy $\boldsymbol{\pi}$ from the theoretical results in single-agent RL (Schulman et al., 2015), which is referred to as the joint TRPO objective,

$$J(\boldsymbol{\pi}_{\text{new}}) - J(\boldsymbol{\pi}_{\text{old}}) \geq \mathcal{L}_{\boldsymbol{\pi}_{\text{old}}}^{\text{joint}}(\boldsymbol{\pi}_{\text{new}}) - C \cdot D_{\text{KL}}^{\max}(\boldsymbol{\pi}_{\text{old}} \| \boldsymbol{\pi}_{\text{new}}) \tag{3}$$

$$\text{where } \mathcal{L}_{\boldsymbol{\pi}_{\text{old}}}^{\text{joint}}(\boldsymbol{\pi}_{\text{new}}) = \sum_s \boldsymbol{\rho}_{\text{old}}(s) \sum_{\boldsymbol{a}} \boldsymbol{\pi}_{\text{new}}(\boldsymbol{a}|s) A_{\text{old}}(s, \boldsymbol{a}), \tag{4}$$

where $D_{\text{KL}}^{\max}(\boldsymbol{\pi}_{\text{old}} \| \boldsymbol{\pi}_{\text{new}}) = \max_s D_{\text{KL}}(\boldsymbol{\pi}_{\text{old}}(\cdot|s) \| \boldsymbol{\pi}_{\text{new}}(\cdot|s))$, $\boldsymbol{\rho}_{\text{old}}(s) = \sum_{t=0} \gamma^t \Pr(s_t = s|\boldsymbol{\pi}_{\text{old}})$ is the discounted stationary distribution of the state given $\boldsymbol{\pi}_{\text{old}}$, $A_{\text{old}}$ is the advantage function under $\boldsymbol{\pi}_{\text{old}}$, and $C$ is a constant.

The joint TRPO objective, *i.e.*, RHS of (3), is a lower bound for the difference between the new joint policy $\boldsymbol{\pi}_{\text{new}}$ and the old joint policy $\boldsymbol{\pi}_{\text{old}}$ in term of expected return. Therefore, we can use this objective as a surrogate, and maximizing this surrogate can guarantee that the policy is improving monotonically. However, the joint TRPO objective cannot be directly optimized in fully decentralized learning as this objective is involved in the joint policy, which cannot be accessed in fully decentralized learning.

We will propose a new lower bound (surrogate) for $J(\boldsymbol{\pi}_{\text{new}}) - J(\boldsymbol{\pi}_{\text{old}})$, which can be optimized in fully decentralized learning. Before introducing our new surrogate, we need to first analyze the critic of agent in fully decentralized learning, which is referred to as decentralized critic.

### 3.2 DECENTRALIZED CRITIC

In fully decentralized learning, each agent learns independently from its own interactions with the environment. Therefore, the Q-function of each agent $i$ is actually the following formula:

$$Q_{\pi^{-i}}^{\pi^i}(s, a_i) = r_{\pi^{-i}}(s, a_i) + \gamma \mathbb{E}_{a_{-i} \sim \pi^{-i}, s' \sim P(\cdot|s, a_i, a_{-i}), a_i' \sim \pi^i}[Q_{\pi^{-i}}^{\pi^i}(s', a_i')], \tag{5}$$

where $r_{\pi^{-i}}(s, a_i) = \mathbb{E}_{\pi^{-i}}[r(s, a_i, a_{-i})]$, and $\pi^{-i}$ and $a_{-i}$ respectively denote the joint policy and joint action of all agents expect agent $i$. If we take the expectation $\mathbb{E}_{a'_{-i} \sim \pi^{-i}(\cdot|s'), a_{-i} \sim \pi^{-i}(\cdot|s)}$ over both sides of the Q-function of joint policy (2), then we have

$$\mathbb{E}_{\pi^{-i}}[Q^{\boldsymbol{\pi}}(s, a_i, a_{-i})] = r_{\pi^{-i}}(s, a_i) + \gamma \mathbb{E}_{a_{-i} \sim \pi^{-i}, s' \sim P(\cdot|s, a_i, a_{-i}), a'_i \sim \pi^i} \left[ \mathbb{E}_{\pi^{-i}}[Q^{\boldsymbol{\pi}}(s', a'_i, a'_{-i})] \right].$$

We can see that $Q^{\pi^i}_{\pi^{-i}}(s, a_i)$ and $\mathbb{E}_{\pi^{-i}}[Q^{\boldsymbol{\pi}}(s, a_i, a_{-i})]$ satisfy the same iteration. Moreover, we will show in the following that $Q^{\pi^i}_{\pi^{-i}}(s, a_i)$ and $\mathbb{E}_{\pi^{-i}}[Q^{\boldsymbol{\pi}}(s, a_i, a_{-i})]$ are just the same.

We first define an operator $\Gamma^{\pi^i}_{\pi^{-i}}$ as follows,

$$\Gamma^{\pi^i}_{\pi^{-i}} Q(s, a_i) = r_{\pi^{-i}}(s, a_i) + \gamma \mathbb{E}_{a_{-i} \sim \pi^{-i}, s' \sim P(\cdot|s, a_i, a_{-i}), a'_i \sim \pi^i}[Q(s', a'_i)].$$

Then we will prove that the operator $\Gamma^{\pi^i}_{\pi^{-i}}$ is a contraction.

Considering any two individual Q-functions $Q_1$ and $Q_2$, we have:

$$
\begin{aligned}
\|\Gamma^{\pi^i}_{\pi^{-i}} Q_1 - \Gamma^{\pi^i}_{\pi^{-i}} Q_2\|_\infty &= \max_{s, a_i} \gamma |\mathbb{E}_{a_{-i} \sim \pi^{-i}, s' \sim P(\cdot|s, a_i, a_{-i}), a'_i \sim \pi^i}[Q_1(s', a'_i) - Q_2(s', a'_i)]| \\
&\le \gamma \mathbb{E}_{a_{-i} \sim \pi^{-i}, s' \sim P(\cdot|s, a_i, a_{-i}), a'_i \sim \pi^i}[\max_{s', a'_i} |Q_1(s', a'_i) - Q_2(s', a'_i)|] \\
&= \gamma \max_{s', a'_i} |Q_1(s', a'_i) - Q_2(s', a'_i)| \\
&= \gamma \|Q_1 - Q_2\|_\infty.
\end{aligned}
$$

So the operator $\Gamma^{\pi^i}_{\pi^{-i}}$ has one and only one fixed point, which means

$$
\begin{aligned}
Q^{\pi^i}_{\pi^{-i}}(s, a_i) &= \mathbb{E}_{\pi^{-i}}[Q^{\boldsymbol{\pi}}(s, a_i, a_{-i})], \\
V^{\pi^i}_{\pi^{-i}}(s) &= \mathbb{E}_{\pi^{-i}}[V^{\boldsymbol{\pi}}(s)] = V^{\boldsymbol{\pi}}(s).
\end{aligned}
$$

With this well-defined decentralized critic, we can further analyze the objective of IPPO (de Witt et al., 2020). In IPPO, the policy objective of each agent $i$ can be essentially formulated as follows:

$$\mathcal{L}^i_{\boldsymbol{\pi}_{\mathrm{old}}}(\pi^i_{\mathrm{new}}) = \sum_s \boldsymbol{\rho}_{\mathrm{old}}(s) \sum_{a_i} \pi^i_{\mathrm{new}}(a_i|s) A^i_{\mathrm{old}}(s, a_i), \tag{6}$$

$$\text{where } A^i_{\mathrm{old}}(s, a_i) = Q^{\pi^i_{\mathrm{old}}}_{\pi^{-i}_{\mathrm{old}}}(s, a_i) - \mathbb{E}_{\pi^i_{\mathrm{old}}}[Q^{\pi^i_{\mathrm{old}}}_{\pi^{-i}_{\mathrm{old}}}(s, a_i)] = \mathbb{E}_{\pi^{-i}_{\mathrm{old}}}[A_{\mathrm{old}}(s, a_i, a_{-i})].$$

However, (6) is different from (4) in the joint TRPO objective. Thus, directly optimizing (6) may not improve the joint policy, and thus cannot provide any guarantee for convergence, to the best of our knowledge. Nevertheless, it seems that $A^i_{\mathrm{old}}(s, a_i)$ is the only advantage formulation that can be accessed by each agent in fully decentralized learning. So, the policy objective of DPO will be derived on (6) but with modifications to guarantee convergence, and we will introduce the detail in the next section. In the following, we discuss how to compute this advantage in practice in fully decentralized learning.

As we need to calculate $A^i_{\mathrm{old}}(s, a_i) = \mathbb{E}_{\pi^{-i}_{\mathrm{old}}}[r(s, a_i, a_{-i}) + \gamma V^{\boldsymbol{\pi}_{\mathrm{old}}}(s') - V^{\boldsymbol{\pi}_{\mathrm{old}}}(s)]$ for the policy update, we can approximate $A^i_{\mathrm{old}}(s, a_i)$ with $\hat{A}^i(s, a_i) = r + \gamma V^{\pi^i}_{\pi^{-i}}(s') - V^{\pi^i}_{\pi^{-i}}(s)$, which is an unbiased estimate of $A^i_{\mathrm{old}}(s, a_i)$, though it may be with a large variance. In practice, we can follow the traditional idea in fully decentralized learning, and let each agent $i$ independently learn an individual value function $V^i(s)$. Then, we further have $\hat{A}^i(s, a_i) \approx r + \gamma V^i(s') - V^i(s)$. The loss for the decentralized critic is as follows:

$$\mathcal{L}^i_{\mathrm{critic}} = \mathbb{E}\left[(V^i(s) - y_i)^2\right], \quad \text{where } y_i = r + \gamma V^i(s') \text{ or Monte Carlo return}. \tag{7}$$

There may be some ways to improve the learning of this critic, which however is beyond the scope of our discussion.

## 3.3 DECENTRALIZED SURROGATE

We are ready to introduce the decentralized surrogate. First, we derive our novel lower bound of the joint policy improvement by the following theorem.

**Theorem 1.** *Suppose $\boldsymbol{\pi}_{\mathrm{old}}$ and $\boldsymbol{\pi}_{\mathrm{new}}$ are two joint policies. Then, the following bound holds:*

$$J(\boldsymbol{\pi}_{\mathrm{new}}) - J(\boldsymbol{\pi}_{\mathrm{old}}) \geq \frac{1}{N} \sum_{i=1}^{N} \mathcal{L}_{\boldsymbol{\pi}_{\mathrm{old}}}^{i}(\pi_{\mathrm{new}}^{i}) - \tilde{M} \cdot \sum_{i=1}^{N} \sqrt{D_{\mathrm{KL}}^{\max}(\pi_{\mathrm{old}}^{i}\|\pi_{\mathrm{new}}^{i})} - C \cdot \sum_{i=1}^{N} D_{\mathrm{KL}}^{\max}(\pi_{\mathrm{old}}^{i}\|\pi_{\mathrm{new}}^{i}),$$

*where $\tilde{M} = \frac{2\max_{s,\boldsymbol{a}}|A_{\mathrm{old}}(s,\boldsymbol{a})|}{1-\gamma}$ and $C = \frac{4\gamma\max_{s,\boldsymbol{a}}|A_{\mathrm{old}}(s,\boldsymbol{a})|}{(1-\gamma)^2}$ are two constants.*

*Proof.* We first consider $\mathcal{L}_{\boldsymbol{\pi}_{\mathrm{old}}}^{\mathrm{joint}}(\boldsymbol{\pi}_{\mathrm{new}}) - \mathcal{L}_{\boldsymbol{\pi}_{\mathrm{old}}}^{i}(\pi_{\mathrm{new}}^{i})$. According to (4) and (6), we have the following equation:

$$
\begin{aligned}
&\mathcal{L}_{\boldsymbol{\pi}_{\mathrm{old}}}^{\mathrm{joint}}(\boldsymbol{\pi}_{\mathrm{new}}) - \mathcal{L}_{\boldsymbol{\pi}_{\mathrm{old}}}^{i}(\pi_{\mathrm{new}}^{i}) \\
&= \sum_{s} \boldsymbol{\rho}_{\mathrm{old}}(s) \sum_{a_i} \pi_{\mathrm{new}}^{i}(a_i|s)\left(\sum_{a_{-i}} \pi_{\mathrm{new}}^{-i}(a_{-i}|s)A_{\mathrm{old}}(s,a_i,a_{-i}) - A_{\mathrm{old}}^{i}(s,a_i)\right) \\
&= \mathbb{E}_{\boldsymbol{\rho}_{\mathrm{old}}}\mathbb{E}_{\pi_{\mathrm{new}}^{i}}\left[\sum_{a_{-i}}\left(\pi_{\mathrm{new}}^{-i}(a_{-i}|s) - \pi_{\mathrm{old}}^{-i}(a_{-i}|s)\right)A_{\mathrm{old}}(s,a_i,a_{-i})\right].
\end{aligned}
$$

Then, we have the following inequalities:

$$
\begin{aligned}
&|\mathcal{L}_{\boldsymbol{\pi}_{\mathrm{old}}}^{\mathrm{joint}}(\boldsymbol{\pi}_{\mathrm{new}}) - \mathcal{L}_{\boldsymbol{\pi}_{\mathrm{old}}}^{i}(\pi_{\mathrm{new}}^{i})| \\
&\leq \mathbb{E}_{\boldsymbol{\rho}_{\mathrm{old}}}\mathbb{E}_{\pi_{\mathrm{new}}^{i}}\left[\sum_{a_{-i}}|\pi_{\mathrm{old}}^{-i}(a_{-i}|s) - \pi_{\mathrm{new}}^{-i}(a_{-i}|s)|\,|A_{\mathrm{old}}(s,a_i,a_{-i})|\right] \\
&\leq \mathbb{E}_{\boldsymbol{\rho}_{\mathrm{old}}}\mathbb{E}_{\pi_{\mathrm{new}}^{i}}\left[M\sum_{a_{-i}}|\pi_{\mathrm{old}}^{-i}(a_{-i}|s) - \pi_{\mathrm{new}}^{-i}(a_{-i}|s)|\right] \qquad (M = \max_{s,\boldsymbol{a}}|A_{\mathrm{old}}(s,\boldsymbol{a})|) \\
&= 2M\mathbb{E}_{\boldsymbol{\rho}_{\mathrm{old}}}\left[D_{\mathrm{TV}}(\pi_{\mathrm{old}}^{-i}(\cdot|s)\|\pi_{\mathrm{new}}^{-i}(\cdot|s))\right] \\
&\leq \frac{2M}{1-\gamma}\max_{s} D_{\mathrm{TV}}(\pi_{\mathrm{old}}^{-i}(\cdot|s)\|\pi_{\mathrm{new}}^{-i}(\cdot|s)) \\
&= \tilde{M}D_{\mathrm{TV}}^{\max}(\pi_{\mathrm{old}}^{-i}\|\pi_{\mathrm{new}}^{-i}) \qquad (\tilde{M} = \frac{2M}{1-\gamma}) \\
&\leq \tilde{M}\sqrt{D_{\mathrm{KL}}^{\max}(\pi_{\mathrm{old}}^{-i}\|\pi_{\mathrm{new}}^{-i})} \hspace{4cm} (8) \\
&\leq \tilde{M}\sqrt{\sum_{j\neq i}D_{\mathrm{KL}}^{\max}(\pi_{\mathrm{old}}^{j}\|\pi_{\mathrm{new}}^{j})}, \hspace{3cm} (9)
\end{aligned}
$$

where (8) is from the relationship between the total variance and KL-divergence that $D_{\mathrm{TV}}(p\|q)^2 \leq D_{\mathrm{KL}}(p\|q)$ (Schulman et al., 2015), and (9) is a property of the KL-divergence, which can be obtained as follows,

$$
\begin{aligned}
D_{\mathrm{KL}}^{\max}(\boldsymbol{\pi}_{\mathrm{old}}\|\boldsymbol{\pi}_{\mathrm{new}}) &= \max_{s} D_{\mathrm{KL}}(\boldsymbol{\pi}_{\mathrm{old}}(\cdot|s)\|\boldsymbol{\pi}_{\mathrm{new}}(\cdot|s)) \\
&= \max_{s}\sum_{i} D_{\mathrm{KL}}(\pi_{\mathrm{old}}^{i}(\cdot|s)\|\pi_{\mathrm{new}}^{i}(\cdot|s)) \\
&\leq \sum_{i}\max_{s} D_{\mathrm{KL}}(\pi_{\mathrm{old}}^{i}(\cdot|s)\|\pi_{\mathrm{new}}^{i}(\cdot|s)) \\
&= \sum_{i} D_{\mathrm{KL}}^{\max}(\pi_{\mathrm{old}}^{i}\|\pi_{\mathrm{new}}^{i}). \hspace{2cm} (10)
\end{aligned}
$$

From (9), we can further obtain the following inequality,

$$\mathcal{L}_{\boldsymbol{\pi}_{\mathrm{old}}}^{\mathrm{joint}}(\boldsymbol{\pi}_{\mathrm{new}}) - \mathcal{L}_{\boldsymbol{\pi}_{\mathrm{old}}}^{i}(\pi_{\mathrm{new}}^{i}) \geq -\tilde{M}\sqrt{\sum_{j\neq i}D_{\mathrm{KL}}^{\max}(\pi_{\mathrm{old}}^{j}\|\pi_{\mathrm{new}}^{j})}. \hspace{2cm} (11)$$

Next we will prove this theorem, starting from (3),

$$J(\boldsymbol{\pi}_{\text{new}}) - J(\boldsymbol{\pi}_{\text{old}}) \geq \mathcal{L}_{\boldsymbol{\pi}_{\text{old}}}^{\text{joint}}(\boldsymbol{\pi}_{\text{new}}) - C \cdot D_{\text{KL}}^{\max}(\boldsymbol{\pi}_{\text{old}}\|\boldsymbol{\pi}_{\text{new}})$$

$$= \frac{1}{N} \sum_{i=1}^{N} \mathcal{L}_{\boldsymbol{\pi}_{\text{old}}}^{\text{joint}}(\boldsymbol{\pi}_{\text{new}}) - C \cdot D_{\text{KL}}^{\max}(\boldsymbol{\pi}_{\text{old}}\|\boldsymbol{\pi}_{\text{new}})$$

$$\geq \frac{1}{N} \sum_{i=1}^{N} \mathcal{L}_{\boldsymbol{\pi}_{\text{old}}}^{i}(\pi_{\text{new}}^{i}) - \frac{\tilde{M}}{N} \sum_{i=1}^{N} \sqrt{\sum_{j \neq i} D_{\text{KL}}^{\max}(\pi_{\text{old}}^{j}\|\pi_{\text{new}}^{j})} - C \cdot D_{\text{KL}}^{\max}(\boldsymbol{\pi}_{\text{new}}\|\boldsymbol{\pi}_{\text{old}}) \qquad (12)$$

$$\geq \frac{1}{N} \sum_{i=1}^{N} \mathcal{L}_{\boldsymbol{\pi}_{\text{old}}}^{i}(\pi_{\text{new}}^{i}) - \tilde{M}\sqrt{\frac{N-1}{N} \sum_{i=1}^{N} D_{\text{KL}}^{\max}(\pi_{\text{old}}^{i}\|\pi_{\text{new}}^{i})} - C \cdot D_{\text{KL}}^{\max}(\boldsymbol{\pi}_{\text{new}}\|\boldsymbol{\pi}_{\text{old}}) \qquad (13)$$

$$\geq \frac{1}{N} \sum_{i=1}^{N} \mathcal{L}_{\boldsymbol{\pi}_{\text{old}}}^{i}(\pi_{\text{new}}^{i}) - \tilde{M}\sqrt{\frac{N-1}{N} \sum_{i=1}^{N} D_{\text{KL}}^{\max}(\pi_{\text{old}}^{i}\|\pi_{\text{new}}^{i})} - C \cdot \sum_{i=1}^{N} D_{\text{KL}}^{\max}(\pi_{\text{old}}^{i}\|\pi_{\text{new}}^{i}) \qquad (14)$$

$$\geq \frac{1}{N} \sum_{i=1}^{N} \mathcal{L}_{\boldsymbol{\pi}_{\text{old}}}^{i}(\pi_{\text{new}}^{i}) - \tilde{M}\sqrt{\sum_{i=1}^{N} D_{\text{KL}}^{\max}(\pi_{\text{old}}^{i}\|\pi_{\text{new}}^{i})} - C \cdot \sum_{i=1}^{N} D_{\text{KL}}^{\max}(\pi_{\text{old}}^{i}\|\pi_{\text{new}}^{i})$$

$$\geq \frac{1}{N} \sum_{i=1}^{N} \mathcal{L}_{\boldsymbol{\pi}_{\text{old}}}^{i}(\pi_{\text{new}}^{i}) - \tilde{M} \cdot \sum_{i=1}^{N} \sqrt{D_{\text{KL}}^{\max}(\pi_{\text{old}}^{i}\|\pi_{\text{new}}^{i})} - C \cdot \sum_{i=1}^{N} D_{\text{KL}}^{\max}(\pi_{\text{old}}^{i}\|\pi_{\text{new}}^{i}). \qquad (15)$$

The inequality (12) is the direct application of the inequality (11). The inequality (13) is from the Cauthy-Schwarz inequality,

$$\sum_{i=1}^{N} \sqrt{\sum_{j \neq i} D_{\text{KL}}^{\max}(\pi_{\text{old}}^{j}\|\pi_{\text{new}}^{j})} \leq \sqrt{N \sum_{i=1}^{N} \sum_{j \neq i} D_{\text{KL}}^{\max}(\pi_{\text{old}}^{j}\|\pi_{\text{new}}^{j})}$$

$$= \sqrt{N(N-1) \sum_{i=1}^{N} D_{\text{KL}}^{\max}(\pi_{\text{old}}^{i}\|\pi_{\text{new}}^{i})}.$$

The inequality (14) is from (10), while the inequality (15) is from the simple inequality $\sqrt{\sum_i a_i} \leq \sum_i \sqrt{a_i}$ ($a_i \geq 0, \forall i$). □

The lower bound in Theorem 1 is dedicated to decentralized policy optimization, because it can be directly decomposed individually for each agent as a decentralized surrogate. From Theorem 1, if we set the policy optimization objective of each agent $i$ as

$$\pi_{\text{new}}^{i} = \arg\max_{\pi^{i}} \left( \frac{1}{N} \mathcal{L}_{\boldsymbol{\pi}_{\text{old}}}^{i}(\pi^{i}) - \tilde{M} \cdot \sqrt{D_{\text{KL}}^{\max}(\pi_{\text{old}}^{i}\|\pi^{i})} - C \cdot D_{\text{KL}}^{\max}(\pi_{\text{old}}^{i}\|\pi^{i}) \right), \qquad (16)$$

then we have $J(\boldsymbol{\pi}_{\text{new}}) \geq J(\boldsymbol{\pi}_{\text{old}})$ from TRPO (Schulman et al., 2015). Moreover, as the objective $J(\boldsymbol{\pi})$ is bounded, the convergence of $\{J(\boldsymbol{\pi}^{t})\}$ is guaranteed, where $\boldsymbol{\pi}^{t}$ is the joint policy after $t$ iterations according to (16). ***Therefore, the joint policy of agents improves monotonically and converges to sub-optimum by fully decentralized policy optimization, i.e., independent learning.*** Note that this result is under the assumption that each agent can obtain the state, and in practice each agent will take the individual trajectory $\tau_i$ as the approximation to the state.

### 3.4 ALGORITHM

DPO is with a simple idea that each agent optimizes the decentralized surrogate (16). However, we face the same trouble as TRPO that the constant $\tilde{M}$ and $C$ are large and if we directly optimize this objective, then the step size of the policy update will be small.

To settle this problem, we absorb the idea of the adaptive coefficient in PPO (Schulman et al., 2017). We use two adaptive coefficients $\beta_1^i$ and $\beta_2^i$ to replace the constant $\tilde{M}$ and $C$ and additionally replace

---

**Algorithm 1. DPO**

---

1: **for** episode = 1 to $M$ **do**
2:     **for** $t = 1$ to max_episode_length **do**
3:         select action $a_i \sim \pi^i(\cdot|s)$
4:         execute action $a_i$ and observe reward $r$ and next state $s'$
5:         collect $\langle s, a_i, r, s' \rangle$
6:     **end for**
7:     Update decentralized critic according to (7)
8:     Update policy according to the surrogate (17)
9:     Update $\beta_1^i$ and $\beta_2^i$ according to (18).
10: **end for**

---

the maximum KL-divergence with the average KL-divergence (Schulman et al., 2015). In practice, we will actually optimize the following objective

$$\pi_{\text{new}}^i = \arg\max_{\pi^i} \left( \frac{1}{N} \mathcal{L}_{\boldsymbol{\pi}_{\text{old}}}^i (\pi^i) - \beta_1^i \sqrt{D_{\text{KL}}^{\text{avg}}(\pi_{\text{old}}^i \| \pi^i)} - \beta_2^i D_{\text{KL}}^{\text{avg}}(\pi_{\text{old}}^i \| \pi^i) \right), \qquad (17)$$

where $D_{\text{KL}}^{\text{avg}}(\pi_{\text{old}}^i \| \pi^i) = \mathbb{E}_{s \sim \boldsymbol{\pi}_{\text{old}}} \left[ D_{\text{KL}}(\pi_{\text{old}}^i(\cdot|s) \| \pi^i(\cdot|s)) \right]$.

As for the adaption of $\beta_1^i$ and $\beta_2^i$, we need to define a hyperparameter $d_{target}$, which can be seen as a ruler for the average KL-divergence $D_{\text{KL}}^{\text{avg}}(\pi_{\text{old}}^i \| \pi_{\text{new}}^i)$ for each agent. If $D_{\text{KL}}^{\text{avg}}(\pi_{\text{old}}^i \| \pi_{\text{new}}^i)$ is close to $d_{target}$, then we believe current $\beta_1^i$ and $\beta_2^i$ are appropriate. If $D_{\text{KL}}^{\text{avg}}(\pi_{\text{old}}^i \| \pi_{\text{new}}^i)$ exceeds $d_{target}$ too much, we believe $\beta_1^i$ and $\beta_2^i$ are small and need to increase and vice versa. In practice, we will use the following rule to update $\beta_1^i$ and $\beta_2^i$:

$$\begin{aligned} &\text{If } D_{\text{KL}}^{\text{avg}}(\pi_{\text{old}}^i \| \pi_{\text{new}}^i) > d_{target} * \delta, \quad \text{then } \beta_j^i \leftarrow \beta_j^i * \omega \quad \forall j \in \{1, 2\} \\ &\text{If } D_{\text{KL}}^{\text{avg}}(\pi_{\text{old}}^i \| \pi_{\text{new}}^i) < d_{target}/\delta, \quad \text{then } \beta_j^i \leftarrow \beta_j^i/\omega \quad \forall j \in \{1, 2\}. \end{aligned} \qquad (18)$$

We choose the constants $\delta = 1.5$ and $\omega = 2$ as the choice in PPO (Schulman et al., 2017). As for the critic, we just follow the standard method in PPO. Then, we can have the fully decentralized learning procedure of DPO for each agent $i$ in Algorithm 1.

The practical algorithm of DPO actually uses some approximations to the decentralized surrogate. Most of these approximations are traditional practice in RL or with no alternative in fully decentralized learning yet. We admit that the practical algorithm may not maintain the theoretical guarantee. However, we need to argue that we go one step further to give a decentralized surrogate in fully decentralized learning with convergence guarantee. We believe and expect that a better practical method can be found based on this objective in future work.

## 4 EXPERIMENTS

In this section, we compare the practical algorithm of DPO with IPPO (de Witt et al., 2020) in a variety of cooperative multi-agent environments, including a cooperative stochastic game, MPE (Lowe et al., 2017), multi-agent MuJoCo (Peng et al., 2021), and SMAC (Samvelyan et al., 2019), covering both discrete and continuous action spaces, and fully and partially observable environments. As we consider fully decentralized learning, in the experiments ***agents do not use parameter-sharing*** as sharing parameters should be considered as centralized learning (Terry et al., 2020). In all experiments, the network architectures and common hyperparameters of DPO and IPPO are the same for a fair comparison. More details about experimental settings and hyperparameters are available in Appendix. Moreover, all the learning curves are from 5 random seeds and the shaded area corresponds to the 95% confidence interval.

### 4.1 A DIDACTIC EXAMPLE

First, we use a cooperative stochastic game as a didactic example. The cooperative stochastic game is with 100 states, 6 agents and each agent has 5 actions. All the agents share a joint reward function. The reward function and the transition probability are both generated randomly. This stochastic

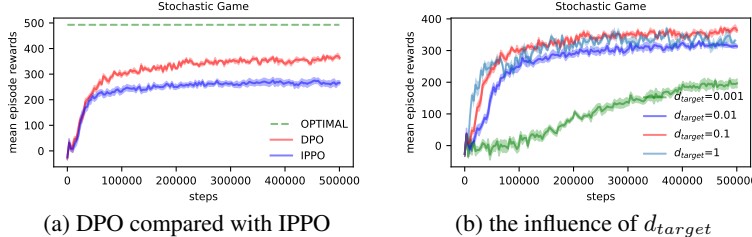

(a) DPO compared with IPPO

(b) the influence of $d_{target}$

Figure 1: Empirical studies of DPO on the didactic example: (a) learning curve of DPO compared with IPPO and the global optimum; (b) the influence of different values of $d_{target}$ on DPO, x-axis is environment steps.

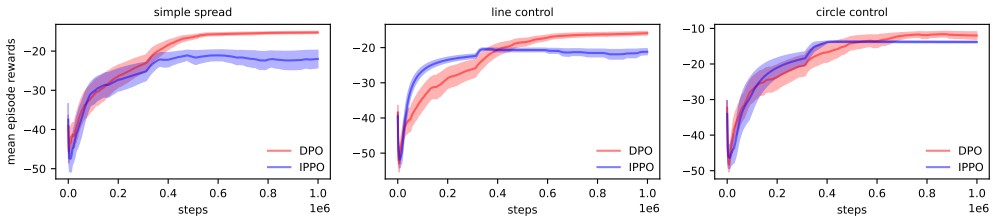

Figure 2: Learning curve of DPO compared with IPPO in 5-agent simple spread, 5-agent line control, and 5-agent circle control in MPE, where x-axis is environment steps.

game has a certain degree of complexity which is helpful to distinguish the performance of DPO and IPPO. On the other hand, this environment is tabular which means training in this environment is fast and we can do ablation studies efficiently. Moreover, we can find the global optimum by dynamic programming to compare with in this game.

The learning curves in Figure 1a show that DPO performs better than IPPO and learns a better solution in this environment. The fact that DPO learns a sub-optimal solution agrees with our theoretical result. However, the sub-optimal solution found by DPO is still away from the global optimum. This means that there is still improvement space.

On the other hand, we study the influence of the hyperparameter $d_{target}$ on DPO. We choose $d_{target} = 0.001, 0.01, 0.1, 1$. The empirical results are shown in Figure 1b. We find that when $d_{target}$ is small, the coefficient $\beta_1$ and $\beta_2$ are more likely to be increased and the step size of the policy update is limited. So for the case that $d_{target} = 0.001, 0.01$, the performance of DPO is relatively low. And when $d_{target}$ is large, the policy update may be out of the trust region. This can be witnessed by the fluctuating learning curve of the case $d_{target} = 1$. So we need to choose an appropriate value for $d_{target}$ and in this environment we choose $d_{target} = 0.1$, which is also the learning curve of DPO in Figure 1a. We found that the appropriate value for $d_{target}$ changes in different environments. In the following, we keep $d_{target}$ to be the same for tasks of the same environment. There may be some better choices for $d_{target}$, but it is a bit time-consuming and out of the range of our discussion.

### 4.2 MPE

MPE is a popular environment in cooperative MARL. MPE is a 2D environment and the objects in MPE environment are either agents or landmarks. Landmark is a part of the environment, while agents can move in any direction. With the relation between agents and landmarks, we can design different tasks. We use the discrete action space version of MPE and the agents can accelerate or decelerate in the direction of x-axis or y-axis. We choose MPE for its partial observability. We take $d_{target} = 0.01$ for all MPE tasks.

The MPE tasks we used for the experiments are simple spread, line control, and circle control which are originally used in Agarwal et al. (2020). In our experiments, we set the number of agents $N = 5$ in all these three tasks. The empirical results are illustrated in Figure 2. We can find that although DPO may fall behind IPPO at the beginning of the training in some tasks, DPO learns a better policy in the end for all three tasks.

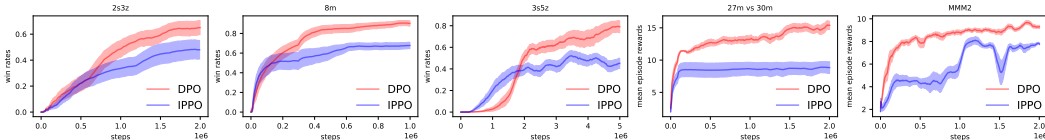

Figure 3: Learning curve of DPO compared with IPPO in 3-agent Hopper, 3-agent HalfCheetah, 3-agent Walker2d, 4-agent Ant, and 17-agent Humanoid in multi-agent MuJoCo, where x-axis is environment steps.

Figure 4: Learning curve of DPO compared with IPPO in 2s3z, 8m, 3s5z, 27m_vs_30m, and MMM2 in SMAC, where x-axis is environment steps.

### 4.3 MULTI-AGENT MUJOCO

Multi-agent MuJoCo is a robotic locomotion control environment for multi-agent settings, which is built upon single-agent MuJoCo (Todorov et al., 2012). In multi-agent MuJoCo, each agent controls one part of a robot to carry out different tasks. We choose this environment for the reason of continuous state and action spaces. We select 5 tasks for our experiments: 3-agent Hopper, 3-agent HalfCheetah, 3-agent Walker2d, 4-agent Ant and 17-agent Humanoid. In all these tasks, we set agent_obsk=2. We take $d_{target} = 0.001$ for all multi-agent MuJoCo tasks.

The empirical results are illustrated in Figure 3. We can find that in all five tasks, DPO outperforms IPPO, though in 3-agent HalfCheetah DPO learns slower than IPPO at the beginning. The results on multi-agent MuJoCo verify that DPO is also effective in facing continuous state and action spaces. Moreover, the better performance of DPO in the 17-agent Humanoid task could be evidence of the scalability of DPO.

### 4.4 SMAC

SMAC is a partially observable and high-dimensional environment that has been used in many cooperative MARL studies. We select five maps in SMAC, 2s3z, 8m, 3s5z, 27m_vs_30m, and MMM2 for our experiments. We take $d_{target} = 0.02$ for all SMAC tasks.

The empirical results are illustrated in Figure 4. The two super hard SMAC tasks (27m_vs_30m and MMM2) are too difficult for both DPO and IPPO to win, so we use episode reward as the metric to show their difference. DPO performs better than IPPO in all five maps. We need to argue that though we have controlled the network architectures of DPO and IPPO to be the same, in our experiments each agent has its individual parameters which increases the difficulty of training. So our results in SMAC may be different from other works. Although IPPO has been shown to perform well in SMAC (de Witt et al., 2020; Yu et al., 2021; Papoudakis et al., 2021), DPO can still outperform IPPO, which verifies the effectiveness of the practical algorithm of DPO in high-dimensional complex tasks and can also be evidence of our theoretical result. Again, the better performance of DPO in 27m_vs_30m shows its good scalability in the task with many agents.

## 5 CONCLUSION

In this paper, we investigate fully decentralized learning in cooperative multi-agent reinforcement learning. We derive a novel decentralized lower bound for the joint policy improvement and we propose DPO, a fully decentralized actor-critic algorithm with convergence guarantee and monotonic improvement. Empirically, we test DPO compared with IPPO in a variety of environments including a cooperative stochastic game, MPE, multi-agent MuJoCo, and SMAC, covering both discrete and continuous action spaces, and fully and partially observable environments. The empirical results show the advantage of DPO over IPPO, which can be evidence for our theoretical results.

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

## A EXPERIMENTAL SETTINGS

### A.1 MPE

The three tasks are built on the origin MPE (Lowe et al., 2017) (MIT license) and are originally used in Agarwal et al. (2020) (MIT license). The objective in these three tasks are listed as follows:

- **Simple Spread:** There are $N$ agents who need to occupy the locations of $N$ landmarks.
- **Line Control:** There are $N$ agents who need to line up between 2 landmarks.
- **Circle Control:** There are $N$ agents who need to form a circle around a landmark.

The reward in these tasks is the distance between all the agents and their target locations. We set the number of agents $N = 5$ for these three tasks in our experiment.

### A.2 MULTI-AGENT MUJOCO

Multi-agent MuJoCo (Peng et al., 2021) (Apache-2.0 license) is a robotic locomotion task with continuous action space for multi-agent settings. The robot could be divided into several parts and each part contains several joints. Agents in this environment control a part of the robot which could be different varieties. So the type of the robot and the assignment of the joints decide a task. For example, the task 'HalfCheetah-3×2' means dividing the robot 'HalfCheetah' into three parts for three agents and each part contains 2 joints.

The details about our experiment settings in multi-agent Mujoco are listed in Table 1. The configuration defines the number of agents and the joints of each agent. The 'agent obsk' defines the number of nearest agents an agent can observe.

Table 1: The task settings of multi-agent MuJoCo

| task | configuration | agent obsk |
|---|---|---|
| HalfCheetah | 3×2 | 2 |
| Hopper | 3×1 | 2 |
| Walker2d | 3×2 | 2 |
| Ant | 4×2 | 2 |

## B TRAINING DETAILS

Our code is based on the open-source code[1] of MAPPO (Yu et al., 2021) (MIT license). We modify the code for individual parameters and ban the tricks used by MAPPO for SMAC. The network architectures and base hyperparameters of DPO and IPPO are the same for all the tasks in all the environments. We use 3-layer MLPs for the actor and the critic and use ReLU as non-linearities. The number of the hidden units of the MLP is 128. We train all the networks with an Adam optimizer. The learning rates of the actor and critic are both 5e-4. The number of epochs for every batch of samples is 15 which is the recommended value in Yu et al. (2021). For IPPO, the clip parameter is 0.2 which is the same as Schulman et al. (2017). For DPO, the initial values of the coefficient $\beta_1^i$ and $\beta_2^i$ are 0.01. The value of $d_{\text{target}}$ is 0.1 for the cooperative stochastic game, 0.01 for MPE, 0.001 for multi-agent MuJoCo, and 0.02 for SMAC.

The version of the game StarCraft2 in SMAC is 4.10 for our experiments in all the SMAC tasks. We set the episode length of all the multi-agent MuJoCo tasks as 1000 in all of our multi-agent MuJoCo experiments. We perform the whole experiment with a total of four NVIDIA A100 GPUs. We have summarized the hyperparameters in Table 2.

---

[1]https://github.com/marlbenchmark/on-policy

Table 2: Hyperparameters for all the experiments

| hyperparameter | value |
|---|---|
| MLP layers | 3 |
| hidden size | 128 |
| non-linear | ReLU |
| optimizer | Adam |
| actor_lr | 5e-4 |
| critic_lr | 5e-4 |
| numbers of epochs | 15 |
| initial $\beta_1^i$ | 0.01 |
| initial $\beta_2^i$ | 0.01 |
| $\delta$ | 1.5 |
| $\omega$ | 2 |
| $d_{\text{target}}$ | different for environments as aforementioned |
| clip parameter for IPPO | 0.2 |

## C  ADDITIONAL RESULTS

Schulman et al. (2017) actually proposed two versions of PPO. The first version, which is also the most popular version, is with the clip trick. The second version is directly optimizing the penalty formula with adaptive coefficients and we refer to this algorithm as PPO-KL. IPPO (de Witt et al., 2020) is actually extended from the first version, while the practical algorithm of DPO is similar to the second version. The main difference between DPO and PPO-KL is the term of the square root of the KL-divergence in the policy loss. We modify IPPO by making each agent learn with PPO-KL to obtain IPPO-KL. On the other hand, independent Q-learning (IQL) (Tan, 1993) is a classic independent learning algorithm. IQL could obtain good performance in some tasks but it is not with any theoretical guarantee, to the best of our knowledge.

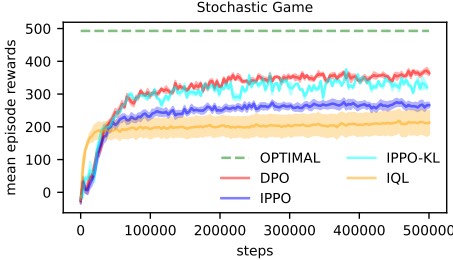

Figure 5: Learning curve of DPO compared with IPPO, IPPO-KL, IQL, and the global optimum in the cooperative stochastic game, where x-axis is environment steps.

For the completeness of our experiments, we test the performance of IPPO-KL and IQL in the cooperative stochastic game and the empirical result is illustrated in Figure 5. We find that the performance of IPPO-KL is close to DPO but a little bit lower and more unstable. This can be explained as the policy loss of IPPO-KL is actually a biased approximation of DPO, which omits the square root term. The theoretical guarantee of DPO may also help IPPO-KL perform better than IPPO in this game, while the bias in the policy loss of IPPO-KL makes it perform worse than DPO. As for IQL, its performance is lower than IPPO though it converges faster. Since IQL performs worse than IPPO in such a didactic environment and IQL is a value-based algorithm and is out of the scope of our discussion, we do not take IQL as a baseline in other environments. Moreover, we think the essential relations among IPPO, IPPO-KL, and DPO can be clearly witnessed from the empirical results in the cooperative stochastic game. For other tasks, we focus on the mainstream version of IPPO as in de Witt et al. (2020); Yu et al. (2021); Papoudakis et al. (2021).

Besides our empirical results, we would like to share our views on the difference between DPO and IPPO and give some intuitive ideas. KL regularization and ratio clipping are similar in the

single-agent setting, but they are not supposed to be similar in multi-agent settings. The 'correct' ratio clipping in multi-agent setting according to the theory of PPO should clip over the joint policy ratio $\frac{\pi_{\mathrm{new}}(\boldsymbol{a}|s)}{\pi_{\mathrm{old}}(\boldsymbol{a}|s)}$. IPPO just clips individual policy ratio $\frac{\pi_{\mathrm{new}}^i(a_i|s)}{\pi_{\mathrm{old}}^i(a_i|s)}$ for each agent $i$ which may not be enough to realize the 'correct' ratio clipping. We could find more discussion about this in the CoPPO (Wu et al., 2021) paper. So IPPO is not supposed to enjoy the theoretical results of DPO.

We could rewrite the objective of IPPO for agent $i$ with a similar formulation in HPO (Yao et al., 2021) as follows:

$$\mathcal{L}_{\boldsymbol{\pi}_{\mathrm{old}}}^{i,\mathrm{IPPO}}(\pi_{\mathrm{new}}^i) = \sum_s \boldsymbol{\rho}_{\mathrm{old}}(s) \sum_{a_i} \pi_{\mathrm{new}}^i(a_i|s)|A_{\mathrm{old}}^i(s,a_i)|l\left(\mathrm{sign}(A_{\mathrm{old}}^i(s,a_i)), u_i(s,a_i)-1, \epsilon\right),$$

where $l(y,x,\epsilon) = \max\{0, \epsilon - y \times x\}$ is the hinge loss and $u_i(s,a_i) = \frac{\pi_{\mathrm{new}}^i(a_i|s)}{\pi_{\mathrm{old}}^i(a_i|s)}$ is the ratio.

If we follow the same idea as PPO, then IPPO is the 'correct' ratio clipping version for the surrogate of DPO. However, the effectiveness of this ratio clipping formulation in theory is still open in decentralized learning since there is not any convergence guarantee for IPPO, to the best of our knowledge.

Though the effectiveness of IPPO in theory is beyond the scope of our paper, we could provide an intuitive explanation for the fact that the performance of DPO can surpass IPPO from this formulation and the analysis in HPO. In the proof of HPO, there is a critical assumption that the sign of the estimated advantage is the same as that of the true advantage (Assumption 4 in Section 2.3 in Yao et al. (2021)). And HPO also shows that the sign of the advantage is more important than the value for this formulation of PPO-clip. In decentralized learning, both DPO and IPPO are facing the difficulty of learning the individual advantage function as there may be noise in the individual value function. However, the objective of DPO is continuous and the objective of IPPO is discrete for $\mathrm{sign}(A_{\mathrm{old}}^i(s,a_i))$. So the impact of the noise in the value function may be larger on IPPO than DPO.

# D DISCUSSION

In the paper, we derive a novel lower bound that can be naturally divided into independent surrogate (16) for each agent. By each agent optimizing this surrogate, the monotonic improvement of the joint policy can be guaranteed in fully decentralized settings. However, the practical algorithm of DPO takes the formula of (17) with several approximations. How to solve the optimization of (16) more precisely is left as future work.

Moreover, we expect our work could provide some insights for future studies on fully decentralized multi-agent reinforcement learning, since current methods still have a gap from the optimum as shown in Figure 5.

# E ADDITIONAL RESULTS FOR REBUTTAL

We first add some experiments results of the ablation study about the objective in (17). We consider two ablation methods: in the first one, we will keep $\beta_1^i = 0$ to eliminate the influence of the term

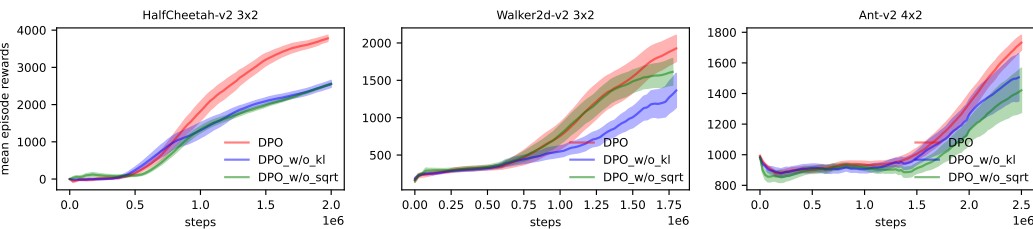

Figure 6: Learning curves of DPO compared with DPO_w/o_sqrt and DPO_w/o_kl in 3-agent HalfCheetah, 3-agent Walker2d, and 4-agent Ant in multi-agent MuJoCo, where x-axis is environment steps.

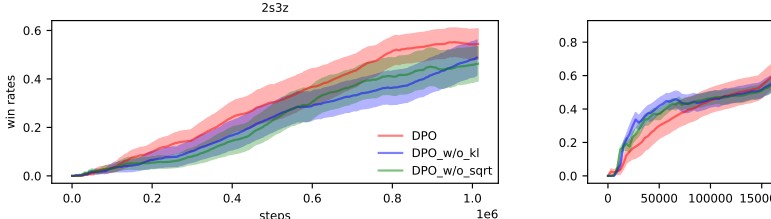

Figure 7: Learning curves of DPO compared with DPO_w/o_sqrt and DPO_w/o_kl on 2s3z and 8m in SMAC, where x-axis is environment steps.

$\sqrt{D_{\mathrm{KL}}^{\mathrm{avg}}(\pi_{\mathrm{old}}^i\|\pi^i)}$, which is actually the same as IPPO-KL; in the second one, we will keep $\beta_2^i = 0$ to eliminate the influence of the term $D_{\mathrm{KL}}^{\mathrm{avg}}(\pi_{\mathrm{old}}^i\|\pi^i)$. The other parameters are controlled to be the same as DPO. We will call these two methods as DPO_w/o_sqrt and DPO_w/o_kl respectively. We select three multi-agent MuJoCo tasks (3-agent HalfCheetah, 3-agent Walker2d and 4-agent Ant) and two SMAC tasks (2s3z and 8m) for this ablation study. The empirical results are included in Figure 6 and Figure 7. We find that eliminating either the term $\sqrt{D_{\mathrm{KL}}^{\mathrm{avg}}(\pi_{\mathrm{old}}^i\|\pi^i)}$ or the term $D_{\mathrm{KL}}^{\mathrm{avg}}(\pi_{\mathrm{old}}^i\|\pi^i)$ will lower the performance of DPO in all these tasks. This could be evidence for the significance of our novel lower bound (16) and the objective (17).

Moreover, we further study the influence the values of $\delta$ and $\omega$ for the adaptive adjustments of the coefficient $\beta_1^i$ and $\beta_2^i$. We test seven different choices of $\delta$ and $\omega$ as $(\delta, \omega) = (1.5, 2), (1.5, 4), (1.5, 6), (1.1, 2), (3, 2), (3, 6), (1.1, 6)$ in the cooperative stochastic game. The empirical results are included in the Figure 8. We find that the influence of different $\delta$ and $\omega$ is relatively limited and the adaptive adjustment is not very sensitive to them. This conclusion is similar to the PPO paper (Schulman et al., 2017).

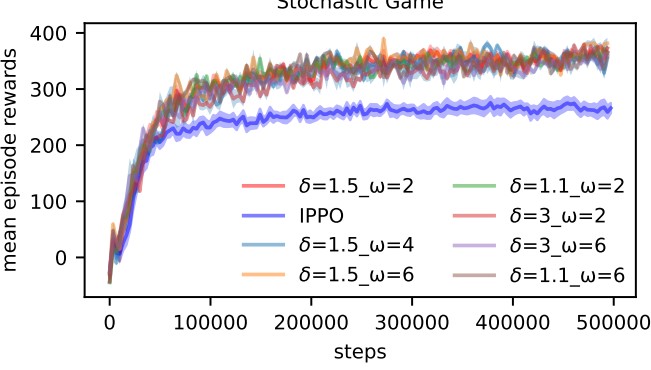

Figure 8: Learning curves of DPO with different values of $\delta$ and $\omega$ in the cooperative stochastic game, where x-axis is environment steps.

