# OpenReview forum: "Decentralized Policy Optimization"
_ICLR.cc/2023/Conference — Submitted to ICLR 2023_

### Official Review · Reviewer_KSta · 2022-10-23

**Confidence:** 4
**Correctness:** 3
**Technical Novelty And Significance:** 2
**Empirical Novelty And Significance:** 2
**Recommendation:** 3

**Clarity, Quality, Novelty And Reproducibility:**

The paper is well-written and easy to follow. However, more technical details may need to be provided regarding some statements in the paper.

The technical novelty of this paper needs to be clearly justified. Specifically, this paper assumes that all learning agents have direct access to the full state information. However, this assumption is often not valid in a multi-agent system.

It remains questionable how the new surrogate is different from the TRPO surrogate and why the difference allows agents to learn to coordinate/cooperate effectively.

The experiment results need to be significantly expanded in order to clearly understand the true advantage of the newly developed algorithm.

**Strength And Weaknesses:**

Strength:
It is important to study new learning mechanisms to support multi-agent reinforcement learning in a decentralized manner. This paper explored an interesting direction of decomposing the trust region surrogate to facilitate independent learning among multiple agents. Experiment results show that this direction of research is promising.

Weakness:
The authors argued that independent learners in a multi-agent system can often outperform agents that adopt CTDE for policy training. However, it remains unclear why independent learners without considering inter-agent interactions can outperform a learning system that explicitly handles inter-agent interactions. Under what conditions will independent learners perform well and why? Besides the potential performance advantage, what are the key benefits of supporting independent learners? The research motivation of this paper may need to be strengthened to answer all these questions.

The paper assumes that all learning agents have direct access to the full state information. However, this assumption is often not valid in a multi-agent system. In fact, with full observability, it is not difficult for each agent to understand the impact of its decision/action on the learning environment and other agents. Therefore, achieving decentralized policy training may not be very difficult. Hence the technical contribution of building a decentralized policy training algorithm under this assumption may need to be justified more. The practical usefulness of the new algorithm also should be investigated more, given the algorithm's critical reliance on the full observability assumption.

The technique for critic training in (7) does not seem to have any difference from critic training in the single-agent setting. Existing research works showed that using a single-agent approach to train critics in a multi-agent setup can be unstable and ineffective under certain conditions, especially when all agents are constantly improving their own policies. It is not clear under what conditions the stability (or convergence) of critic learning can be guaranteed and whether those conditions are realistic for multi-agent reinforcement learning.

Regarding the new lower bound for each agent in (16), the middle term with $\sqrt{D^{max}_{KL}}$ can be simply removed by adjusting $C$ in the third term. Given that, the new surrogate in DPO is essentially the same as the TRPO surrogate for single-agent reinforcement learning. As a result, I can hardly understand how the new surrogate, as a result of including a removable middle term, can facilitate effective inter-agent coordination/cooperation. It also remains questionable how DPO is fundamentally different from a direct extension of TRPO to the multi-agent setup. What are the technical advantages of DPO, in comparison to independent TRPO agents?

The new DPO algorithm introduces several new hyper-parameters such as $d_{target}$, $\delta$, $\omega$, $\beta_1^i$ and $\beta_2^i$, that may need to be fine-tuned. The introduction of these hyper-parameters may make it difficult to apply DPO to a wide range of problems.

More technical details may need to be provided regarding some statements in the paper. For example, it is stated on page 7 that "DPO actually some approximations to the decentralized surrogate". What approximations does this statement refer to?

The authors argued that DPO can be guaranteed to converge. However, I cannot find a thorough analysis on the convergence properties of DPO. In my opinion, a high-level argument regarding the adoption of the trust-region surrogate is not sufficient to show the convergence of the DPO algorithm. Perhaps the claim on algorithm convergence should be adjusted in the paper.

While the experiment results appear promising, DPO is only compared with IPPO in the experiments. It is not clear whether DPO can outperform state-of-the-art CTDE algorithms on any of the benchmark problems. It also remains unclear whether DPO can outperform independent TRPO agents. Hence it is questionable whether the observed performance difference is due to the use of TRPO for policy training or due to the development of the new decentralized surrogate.


**Summary Of The Paper:**

This paper investigated the possibility of decomposing the trust region surrogate over joint policies into the respective surrogates for each individual agent's policies for cooperative multi-agent reinforcement learning. The reported experiment results show that the newly developed DPO algorithm based on the decomposed surrogates can outperform a multi-agent system with independent PPO learners.

**Summary Of The Review:**

It is important to study new learning mechanisms to support multi-agent reinforcement learning in a decentralized manner. However, the key assumption of this paper seems to have some major limitations. The technical novelty of the newly developed performance bound also remains questionable to a certain extent. Furthermore, experimental study is at the limited side and does not convincingly show the true advantage of the newly developed algorithm.

---

> ### Author Response · Authors · 2022-11-18
> **Reply to Reviewer KSta part2**
>
> > More technical details may need to be provided regarding some statements in the paper. For example, it is stated on page 7 that "DPO actually some approximations to the decentralized surrogate". What approximations does this statement refer to?
>
>
>
>
>
> The approximations in this statement refer to the practical method of DPO in Section 3.4 such as replacing the constant $C$ and $\tilde{M}$ with adaptively coefficient $\beta^i_1$ and $\beta^i_2$ and replacing the maximum KL-divergence with the average KL-divergence.
>
>
>
>
>
> > The authors argued that DPO can be guaranteed to converge. However, I cannot find a thorough analysis on the convergence properties of DPO.
>
>
>
>
>
> We have stated the convergence after Theorem 1 in Section 3.3. By optimizing the objective in (16), we could obtain a joint policy sequence $\{J(\boldsymbol{\pi}^t)\}$ which improves monotonically. Then as this sequence is bounded, so the convergence is guaranteed. Our main contribution is the novel lower bound which could be optimized by each agent independently. After we obtain this lower bound, the proof of the convergence is just following the idea of TRPO and is quite simple. We think we have stated this result clearly enough.
>
>
>
>
>
> > While the experiment results appear promising, DPO is only compared with IPPO in the experiments. It is not clear whether DPO can outperform state-of-the-art CTDE algorithms on any of the benchmark problems. It also remains unclear whether DPO can outperform independent TRPO agents. Hence it is questionable whether the observed performance difference is due to the use of TRPO for policy training or due to the development of the new decentralized surrogate.
>
>
>
>
>
> We do not understand the reason for comparing DPO with CTDE methods. The fully decentralized setting is different from CTDE and more general. We think fully decentralized setting is valuable and independent learning is the solution to this setting. So we propose an independent learning method with convergence guarantee and our purpose is not to find a simple way to approximate the performance of SOTA CTDE methods.
>
>
>
> As for the independent TRPO agents, we think PPO is thought as a simpler but still effective version of TRPO in single agent RL and IPPO has shown its performance in multi-agent RL. So we think it is enough to compare DPO with IPPO.
>
>
>
> See the summary for the content about extra experiments.

---

> ### Author Response · Authors · 2022-11-18
> **Reply to Reviewer KSta part1**
>
> > The authors argued that independent learners in a multi-agent system can often outperform agents that adopt CTDE for policy training. However, it remains unclear why independent learners without considering inter-agent interactions can outperform a learning system that explicitly handles inter-agent interactions. Under what conditions will independent learners perform well and why? Besides the potential performance advantage, what are the key benefits of supporting independent learners? The research motivation of this paper may need to be strengthened to answer all these questions.
>
>
>
>
>
> The questions you mentioned are important for this research direction, however, answering all these questions in one paper is impossible and beyond our ability. IPPO outperforms some CTDE methods is empirically verified in previous works such as Papoudakis et al., 2021. Answering this question is not our focus but should attract attention from the MARL community. However, the community just pays too much attention to CTDE. There are many settings CTDE cannot address, for example, in scenarios where no good simulator is available or communication during training is strictly limited.
>
>
>
> Our contribution is that we show that independent learners are able to guarantee the convergence of the joint policy and the non-stationarity caused by simultaneous policy updates of independent learners could be tackled by our independent policy objectives. We think this contribution is very meaningful to independent learning.
>
>
>
>
>
> > The paper assumes that all learning agents have direct access to the full state information.
>
>
>
> See the summary.
>
>
>
>
>
> > The technique for critic training in (7) does not seem to have any difference from critic training in the single-agent setting. Existing research works showed that using a single-agent approach to train critics in a multi-agent setup can be unstable and ineffective under certain conditions, especially when all agents are constantly improving their own policies.
>
>
>
>
>
> For the decentralized critic, the only condition could be used by the independent agent may be that $\mathbb{E}[r_i(s,a_i)] = \mathbb{E}\_{\pi_{-i}}[r(s,a_i,a_{-i})]$, so we will train the decentralized critic $Q_i^{\boldsymbol{\pi}}(s,a_i)$ with the on-policy data of the joint policy $\boldsymbol{\pi}$, this is an unbiased estimate. So under a condition such as we have enough on-policy data, this training is correct. Moreover, we have mentioned in Section 3.2 that improving the learning of the critic is beyond the scope of this paper.
>
>
>
>
>
> > About removing the square root term.
>
>
>
> See the summary.
>
>
>
>
>
> > About hyper-parameters.
>
>
>
> Actually, $\beta^i_1$ and $\beta^i_2$ are not hyper-parameters, they are adjusted adaptively in the training. And the value of $\delta$ and $\omega$ is taken from PPO which states that the performance is not very sensitive to the value of $\delta$ and $\omega$ . So the main hyper-parameter is actually only $d_{\operatorname{target}}$ and we have studied the influence of $d_{\operatorname{target}}$ in the cooperative stochastic game and chose an appropriate value for $d_{\operatorname{target}}$ in other tasks. Adjusting the value of $d_{\operatorname{target}}$ may improve the performance but it is beyond our scope of discussion.
>
>
>
> See the summary for the content about extra experiments.

---

### Official Review · Reviewer_byTG · 2022-10-24

**Confidence:** 4
**Correctness:** 3
**Technical Novelty And Significance:** 2
**Empirical Novelty And Significance:** 2
**Recommendation:** 3

**Clarity, Quality, Novelty And Reproducibility:**

\
<Novelty and Quality>

The implementation part mainly relies on the techniques of TRPO and PPO.

- In my opinion, citing TRPO is required to “replace the maximum KL-divergence with the average KL-divergence.” In addition, using the adaptive coefficient technique requires citation of PPO, not to mention the value of $\delta$ and $\omega$.
- The PPO paper states that PPO “is not very sensitive to” the value of $\delta$ and $\omega$. DPO also requires the same sensitivity check as a multi-agent setting can be more unstable than a single-agent case due to the non-stationary issue.


<Reproducibility>

Since the authors did not provide their source code, it was hard to check the reproducibility of the experiments and several experimental details.


<Clarity>

Many parts of the paper should be clarified. Please refer to the detailed questions below.

1. This paper states that “the network architectures and common hyperparameters of DPO and IPPO are the same for a fair comparison.” (page 6) While the original IPPO paper uses Generalized Advantage Estimation (GAE), DPO does not use GAE and uses the standard TD error for advantage estimation (stated on page 4).
- Is there any specific reason why DPO does not use GAE?
- Does the compared IPPO in the experiments use GAE? Does IPPO use value clipping and the entropy term?
- For clarity, DPO does not use value clipping and the entropy term, does it?


2. Regarding the proposed DPO loss, further ablation study is required.
Naively thinking, the second and the third term in Eq. 17 may have a similar role. What if we do $\beta^i_1 = 0$ or $\beta^i_2 = 0$?
- If $\beta^i_1 = 0$, it produces IPPO-KL in Appendix C. I think the other environments also require IPPO-KL as a baseline for a fair comparison. If the gap between DPO and IPPO-KL is small, the effect of the proposed method may weaken.
- If $\beta^i_2 = 0$, we may still consider trust-region due to the square root term. A related ablation study is required.


3. Can we use more tighter bound for implementation, e.g., Eq.14? Although Eq.14 is not agent-wise separable, we can easily optimize it with modern deep-learning libraries. If the gap between the true loss and the lower bound becomes larger, the learning efficiency becomes worse. Experiments using tighter bound are required to support the necessity of Eq.15, i.e., less-tight but agent-wise separable loss.

4. The legends of Figures 3 and 4 should be larger than the current ones. It is hard to read the details in each graph. In addition, several readers may overlook that authors reported average reward instead of win rate in ‘27m vs 30m’ and ‘MMM2’. This part should also be explicitly and more vividly described in the paper for the readers.

5.  ‘agent_obsk=2’ should be explicitly stated in the main paper (not only in the Appendix) for readers.

6. In the second line on page 4, it seems that $\mathbb{E}_{a_{-i}' \sim \pi^{-i}(\cdot|s’)}$ may be redundant.

7. The claim that "the better performance of DPO in the 17-agent Humanoid task could be evidence of the scalability of DPO" (page 9) can be better supported if there is any experiment showing a marginal performance gap. (The improvement ratio is the smallest among the 5 figures in Fig. 3.)

8. What is the SC2 version used?

9. For clarity, does the DPO use synchronized sampling (i.e., each agent samples the same episodes), or does each agent sample independently given an (either on-policy or off-policy) batch data?




<Discussion>

- The fine-tuned value of $d_{target}$ becomes lower as the environment becomes more complex. Could the authors provide any discussion regarding this observation?
- Could the authors provide any discussion why there is a reward drop at the 1.5M steps in MMM2?





**Strength And Weaknesses:**

Pros: The authors dealt with a theoretical derivation of the suggested surrogate loss.

Cons: Several parts of the paper should be clarified, and additional experiments should be performed. It was hard to check the reproducibility. Please refer to the detailed questions below.



**Summary Of The Paper:**

This paper proposes a fully decentralized policy optimization (DPO) algorithm in cooperative multi-agent reinforcement learning (MARL). Based on the surrogate loss of the trust-region policy optimization for the joint policy, the authors derive a lower bound of the joint policy improvement under the assumption that each agent can obtain the state, and joint policy is represented as the product of individual policy. This agent-wise separable loss is optimized using two adaptive coefficients to overcome small step sizes in a naive application. Experiments in several cooperative MARL environments show that the proposed DPO can perform better than independent PPO under fully decentralized settings.


**Summary Of The Review:**

Although the authors provided theoretical insight into policy optimization in a fully decentralized MARL setting, the novelty and reproducibility are not well supported. In addition, several parts of the paper should be clarified by performing additional experiments.

---

> ### Author Response · Authors · 2022-11-18
> **Reply to Reviewer byTG**
>
> > About GAE, value clipping and entropy term
>
>
>
> In our statement about the TD error, we just provide a way to update the critic. Except for TD error, the other reasonable method for the critic update is also allowed. We actually take GAE, value clipping and entropy term as a part of the architecture of IPPO, so in practice, both IPPO and DPO use these tricks. We will modify our statements to make this clear.
>
>
>
>
>
> > About the ablation of $\beta^i_1$ and $\beta^i_2$
>
>
>
> See the summary.
>
>
>
>
>
> > Can we use more tighter bound for implementation?
>
>
>
> For the fully decentralized setting, agents need agent-wise loss and this is the necessity for Eq.15. Our purpose is to provide an independent learning method for the fully decentralized setting with convergence guarantee. If we just abandon the fully decentralized setting, optimizing the objective such as Eq.3 with centralized method may be even better, but this is not what we want.
>
>
>
>
>
> > About the figure and the statement of agent\_obsk.
>
>
>
> We will adjust this in the revision.
>
>
>
>
>
> > About the second line in page 4
>
>
>
> We mean that the expectation is over both the current step action $a_{-i}$ and the next step action $a_{-i}^\prime$ .
>
>
>
>
>
> > About the SC2 version
>
>
>
> All SMAC experiments are with SC2 version 4.6.
>
>
>
>
>
> > About the sample
>
>
>
> Each agent samples independently.
>
>
>
> > About the scalability of DPO
>
>
>
> We think our results on the tasks of Humanoid-v2\_17x1 and 27m\_vs\_30m is enough to support our claim.
>
>
>
> > About the citation of TRPO and PPO
>
>
>
> We will add the corresponding citation in Section 3.4 in the revision.
>
>
>
> > About the influence of $\delta$ and $\omega$
>
>
>
> See the summary.
>
>
>
>
>
> > Could the authors provide any discussion why there is a reward drop at the 1.5M steps in MMM2?
>
>
>
> We have checked the data and found that the learning curve of one seed dropped at around 1.5M steps. This may be fortuity caused by random seeds.
>
>
>
>
>
> >  The fine-tuned value of $d_{target}$ becomes lower as the environment becomes more complex. Could the authors provide any discussion regarding this observation?
>
>
>
> The multi-agent MuJoCo environment is with continuous action space so the KL calculation is different from the other three discrete action space environments which may be the reason that the corresponding $d_{target}$ is the lowest. As for the other three environments, we believe that the difficulties are ranked as stochastic game < MPE < SMAC, but the corresponding value 0.2, 0.01, 0.02 seems to be not ranked. We prefer to believe that there may not be much regularity between the value of $d_{target}$ and the difficulty as we do not take a detailed search in MPE, SMAC and multi-agent MuJoCo.

---

### Official Review · Reviewer_9jhx · 2022-10-24

**Confidence:** 5
**Correctness:** 2
**Technical Novelty And Significance:** 2
**Empirical Novelty And Significance:** 2
**Recommendation:** 3

**Clarity, Quality, Novelty And Reproducibility:**

The paper is generally easy to follow, with a clear introduction, a rounded discussion of related work, and a well-motivated method. The final policy optimization objective of each agent with two regularization terms is somewhat interesting. However, more discussion and experiments are needed to show the necessity of using two regularization terms.

**Strength And Weaknesses:**

**Strengths**
1. The paper proposes a well-motivated technique for monotonic improvement in multi-agent optimization based on decentralized learning. The derivation is clear and easy to follow.
2. The proposed method and IPPO are compared in multiple multi-agent environments, including MPE, multi-agent MuJoCo, and SMAC.

**Weaknesses**
1. The experiment is not sufficient enough to convince the benefits of the proposed method.
   a) The details of IPPO are not mentioned. To limit the KL divergence between the last and updated policies, one can use a clip function or an adaption of beta as DPO. Which one is used in this paper?
   b) Fig.1 (b) shows that d_target dramatically influences the final performance. The performance of DPO with the worst hyper-parameter is worse than IPPO. If IPPO uses an adaption of beta as DPO, will its performance become better? Is IPPO’s best d_target the same as that in DPO?
   c) The hyper-parameter d_target is changed a lot across different domains. Is there not a single parameter configuration that could perform well in all environments? More studies are required to show the sensitivity of this hyper-parameter in different multi-agent environments.
   d) Following c), fine-tuning d_target might lead to a dramatically different performance in various environments. The reviewer wonders whether IPPO is also fine-tuned for a fair comparison.

2. The paper proposes a new policy optimization objective for each agent as shown in Eq. 16 based on the lower bound introduced in Eq. 15. However, the sqrt of the KL divergence in Eq. 15 can be easily replaced by the KL divergence itself (for example, based on a piecewise function with a different weight) for a further lower bound. Based on this further lower bound, the policy optimization objective of each agent will be similar to that in IPPO. Thus, it is quite important for the paper to show more experiments and details, as mentioned above, to clarify the importance of explicitly optimizing the sqrt of the KL divergence between the last and updated policies.

3. The paper holds an assumption for the theoretical guarantee that each agent could receive the state. This is reasonable because the analysis in partially observable environments is much more difficult, and the problem may be undecidable in Dec-POMDP. However, for decentralized learning, one crucial factor influencing agents' cooperation is incomplete information. It will be helpful to discuss the influence of partial observation on theoretical analysis.

4. This paper does not discuss the limitations of the proposed method.




**Summary Of The Paper:**

Based on the setting of decentralized learning, the paper proposes a decentralized actor-critic algorithm with monotonic improvement and convergence guarantee, where each agent independently optimizes the surrogate. The experiment results show that the proposed algorithm DPO outperforms IPPO in most tasks.

**Summary Of The Review:**

Based on the assumption that each agent could receive the state, the paper proposes a new optimization objective to guarantee the whole team's monotonic improvement with an sqrt of the KL divergence between the last and updated policies. However, further discussion and experiments are required to show the necessity of the new optimization term based on a fair comparison with IPPO. Meanwhile, further analysis of the influence of partial observation in both theoretical and experimental parts is necessary. Therefore, the paper requires a major revision.

---

> ### Author Response · Authors · 2022-11-18
> **Reply to Reviewer 9jhx**
>
> > The details of IPPO are not mentioned.
>
>
>
> We have cited the IPPO paper and stated that the network architectures and common hyperparameters of DPO and IPPO are the same for a fair comparison. So the IPPO in our paper uses the clip function.
>
>
>
>
>
> > If IPPO uses an adaption of beta as DPO, will its performance become better? Is IPPO’s best d_target the same as that in DPO?
>
>
>
> We call this version of IPPO as IPPO-KL and we compare IPPO-KL with DPO in the stochastic game in Appendix B. See the summary for the content about extra experiments.
>
>
>
> >The hyper-parameter d_target is changed a lot across different domains. Is there not a single parameter configuration that could perform well in all environments? More studies are required to show the sensitivity of this hyper-parameter in different multi-agent environments.
>
>
>
> The multi-agent MuJoCo environment is with continuous action space so the KL calculation is different from the other three discrete action space environments, so it is a special case. As for the other environments, if we just need to perform well (better than baseline), then the $d_{target} = 0.01$ is a choice.
>
>
>
>
>
> > The reviewer wonders whether IPPO is also fine-tuned for a fair comparison.
>
>
>
> We use the open-source code of IPPO from the MAPPO paper and take the parameter as the paper suggests. The original paper has done a detailed hyper-parameter search.
>
>
>
>
>
> > About the square root term.
>
>
>
> See the summary.
>
>
>
>
>
> > About the assumption of the full state information.
>
>
>
> See the summary.
>
>
>
>
>
> > About the limitation.
>
>
>
> We have discussed the limitation of our method in Appendix D. The limitation includes that the practical algorithm of DPO takes the formula of equation 17 with several approximations and DPO is on-policy which may result in difficulties in sample efficiency.

---

### Author Response · Authors · 2022-11-18
**Summary**

We would like to reply to some common questions here.





> About the square root term.



We argue that **removing the term $\sqrt{D_{\operatorname{KL}}^{\operatorname{max}} }$ is not as easy as some reviewers say**. There are two main difficulties: 1) If we use the inequality $\sqrt{x} \le \frac{1}{2} \left( u + \frac{x}{u} \right) \quad \forall u \in \mathbb{R}^+$ , then there will be an extra term $-\tilde{M}u$ in the lower bound in (15), so you need to prove that this lower bound is a surrogate, *e.g.*, the maximum of this expression is larger than zero and the condition when the equality is achieved for the inequality $\sqrt{x} \le \frac{1}{2} \left( u + \frac{x}{u} \right)$ is different with the other inequalities; 2) $\sqrt{D_{\operatorname{KL}}^{\operatorname{max}}} \ge D_{\operatorname{KL}}^{\operatorname{max}}$ when $D_{\operatorname{KL}}^{\operatorname{max}} \le 1$.



We would be very pleased if the reviewers could provide some simple methods to eliminate the square root term and we will modify our deduction according to such methods.





> About the assumption of the full state information.



The assumption of the full state information is mostly for the theoretical analysis. We do not know any work with convergence analysis in partially observable environments without such an assumption. Even CTDE methods such as value decomposition do have such an assumption. In our opinion, without such an assumption, convergence analysis is infeasible. We also have mentioned in the preliminaries section that existing works have already shown the difficulty of Dec-POMDP.



Moreover, our main contribution is to tackle the non-stationarity for independent learners. The convergence analysis in partially observable environments would be great but beyond the scope of our discussion. Even though some reviewers say that achieving decentralized policy training may not be very difficult, we think our result is the first independent policy-based method with convergence guarantee and simultaneous policy updates in such a setting, to the best of our knowledge. Finally, our empirical results include partially observable environments, which could be evidence for the practical usefulness of DPO.



> About the ablation study of $\beta_1^i$ and $\beta_2^i$.

We have added experiments in multi-agent MuJoCo and SMAC tasks about eliminating the term $\sqrt{D^{\operatorname{avg}}\_{\operatorname{KL}}(\pi\_{\operatorname{old}}^{i} \Vert \pi^{i} )}$ (keep $\beta^i_1 = 0$)  or the term $D^{\operatorname{avg}}\_{\operatorname{KL}}(\pi_{\operatorname{old}}^{i} \Vert \pi^{i} )$ (keep $\beta^i_2 = 0$)  in the objective (17) in Appendix E. The empirical results are included in the Figure 6 and Figure 7. We find that eliminating either the term $D^{\operatorname{avg}}\_{\operatorname{KL}}(\pi_{\operatorname{old}}^{i} \Vert \pi^{i} )$ or the term $\sqrt{D^{\operatorname{avg}}\_{\operatorname{KL}}(\pi_{\operatorname{old}}^{i} \Vert \pi^{i} )}$ will lower the performance of DPO in all these tasks.  This could be evidence for the significance of our novel lower bound (16) and the objective (17).



**Due to the time limit, some experiments are still running and we will update the empirical results as soon as possible.**



> About the influence of $\delta$ and $\omega$.





We have added experiments in the cooperative stochastic game about the influence of different choices of $\delta$ and $\omega$ in Appendix E. The empirical results are included in the Figure 8. We could find that the influence of different $\delta$ and $\omega$ is relatively limited and the adaptive adjustment is not very sensitive to them.

---

> ### Comment · Reviewer_KSta · 2022-11-18
> **Regarding the use of sqrt in the bound**
>
> Thank the authors for providing some explanations about the importance of using the sqrt term in the surrogate/bound. Looking at eq. (17) which involves the use of this sqrt term, if the aim is to maximize this surrogate by gradient descent/ascent, the gradient for the KL part will assume the form of $(-\frac{\beta_1}{2\sqrt{D_{KL}^{Avg}}}-\beta_2)\nabla D_{KL}^{Avg}$. When $D_{KL}^{Avg} \approx 0$, $(-\frac{\beta_1}{2\sqrt{D_{KL}^{Avg}}}-\beta_2)$ may approach to $-\infty$. Why is this important for multi-agent reinforcement learning? Why will such policy training be stable? Is this risk somehow addressed by exponentially decreasing $\beta_1$ and $\beta_2$ in (18)? However, for on-policy learning, wouldn't $D_{KL}^{Avg}\approx 0$ initially? What is the initial value of $\beta_1$ that can ensure stability of policy training?
>
> On the other hand, using the inequality provided by the authors, $\sqrt{D_{KL}^{Avg}}\leq \frac{1}{2}(u+D_{KL}^{Avg}/u)$, can we re-write the KL part in eq. (17) as $-\frac{\beta_1}{2}(u+D_{KL}^{Avg}/u)-\beta_2 D_{KL}^{Avg}$? The corresponding gradient would become $-\frac{\beta_1}{2u}\nabla D_{KL}^{Avg}-\beta_2 \nabla D_{KL}^{Avg}=-\beta_3(u)\nabla D_{KL}^{Avg}$, where we can simply make $\beta_3(u)$ adaptive to enhance performance. Policy training is also stable even when $D_{KL}^{Avg} \approx 0$. Why is this a bad idea in comparison to keeping $\sqrt{D_{KL}^{Avg}}$ explicitly in the surrogate?
>
> In addition to the above, I probably need more information to understand why $\sqrt{D_{KL}^{Avg}}$ in eq. (17) is particularly important for multi-agent rather than single-agent reinforcement learning, if what it means algorithmically is to adaptively change $\beta_3$ based on $D_{KL}^{Avg}$.

---

> > ### Author Response · Authors · 2022-11-18
> > **Reply**
> >
> > > About the gradient of $\sqrt{D_{\operatorname{KL}}^{\operatorname{max}} }$.
> >
> >
> >
> > We would like to believe that the calculation of the gradient of objective (17) is a part of the practical method. The unstable problem does exist in our experiments, so we use $\sqrt{D_{\operatorname{KL}}^{\operatorname{max}} + \epsilon }$ to approximate this term where we take $\epsilon = 1e-12$.  That's why we actually do not like this term and we have tried some methods to eliminate this term in theory but failed.
> >
> >
> >
> > > About the inequality $\sqrt{x} \le \frac{1}{2} \left( u + \frac{x}{u} \right)$.
> >
> >
> >
> > This inequality is actually one of our failed attempts to eliminate the sqrt term. **This method is not proven and may not be corrected.** If we use this inequality, then we need to prove that $\max_{\pi^i} \Big(\frac{1}{N}\mathcal{L}^{i}\_{\boldsymbol{\pi}\_{\operatorname{old}}}(\pi^i) - (C + \frac{\tilde{M}}{u}) \cdot D^{\max}\_{\operatorname{KL}}(\pi\_{\operatorname{old}}^{i} \Vert \pi^i) - \tilde{M} u \Big) \ge 0$ or find some $\tilde{\pi}^i$ make sure that $\frac{1}{N}\mathcal{L}^{i}\_{\boldsymbol{\pi}\_{\operatorname{old}}}(\tilde{\pi}^i) - (C + \frac{\tilde{M}}{u}) \cdot D^{\max}\_{\operatorname{KL}}(\pi\_{\operatorname{old}}^{i} \Vert \tilde{\pi}^i)  \ge \tilde{M} u$. If this property is not satisfied then the monotonic improvement of the joint policy is not guaranteed and the other theoretical results such as the convergence are not supported. This is the critical problem before we discuss the calculation and property of the gradient. Unfortunately, we haven't settled this problem and we will be pleased if anyone could provide some feasible ideas.  But the same problem for objective (17) is easy since we could just take $\pi^i = \pi_{\operatorname{old}}^{i}$ to make sure the expression equals to 0.
> >
> >
> >
> > > About the importance of the term $\sqrt{D_{\operatorname{KL}}^{\operatorname{max}} }$ in multi-agent settings.
> >
> >
> >
> > If you want some explanations or examples about how this term helps agents to coordinate in multi-agent settings, we may disappoint you. **This term just arises in our deductions in Theorem 1** and we just know optimizing the objective (17) could make sure the joint policy improves monotonically which could further guarantee the convergence of the joint policy.  The importance of the term $\sqrt{D_{\operatorname{KL}}^{\operatorname{max}} }$ may lie in inequality (11) where this term connects the joint policy objective and the individual policy objective.  Actually, in single-agent settings,  inequality (11) is not needed since the two terms in LHS are exactly the same.

---

> > > ### Comment · Reviewer_KSta · 2022-11-18
> > > **Reply**
> > >
> > > Thank you for further clarification. When you change $\sqrt{D_{KL}^{max}}$ to $\sqrt{D_{KL}^{max}+\epsilon}$, eq. (16) also cannot be guaranteed to be positive (at least in theory). When $D_{KL}^{max}\approx 0$, the gradient can still be unstable. I guess it takes additional efforts to address this problem in eq. (16).

---

> > > > ### Author Response · Authors · 2022-11-19
> > > > **Reply**
> > > >
> > > > There may be some misunderstandings for the term $\sqrt{D_{\operatorname{KL}}^{\operatorname{avg}} + \epsilon }$. We mean that we will replace $\sqrt{D_{\operatorname{KL}}^{\operatorname{avg}} }$ with  $\sqrt{D_{\operatorname{KL}}^{\operatorname{avg}} + \epsilon }$ in optimizing the objective (17) but in the deductions for surrogate (16) we will use the exact $\sqrt{D_{\operatorname{KL}}^{\operatorname{max}}}$.
> > > >
> > > >
> > > >
> > > > We have admitted that our practical method uses some approximations to optimize the decentralized surrogate in Section 3.4 and we also have admitted that how to solve the optimization of (16) more precisely is left as future work in Appendix D.  Replacing $\sqrt{D_{\operatorname{KL}}^{\operatorname{avg}} }$ with $\sqrt{D_{\operatorname{KL}}^{\operatorname{avg}} + \epsilon }$ is an approximation to settle the difficulty in practical calculation and is not related to the correctness of surrogate (16). Moreover, our empirical results have verified the effectiveness of our method and our ablation study has showed the significance of the term $\sqrt{D_{\operatorname{KL}}^{\operatorname{max}} }$ in the objective.

---

### Decision · Program_Chairs · 2023-01-20

**Decision:**

Reject

**Justification For Why Not Higher Score:**

Preliminary results not yet ready for publication

**Justification For Why Not Lower Score:**

N/A

**Metareview: Summary, Strengths And Weaknesses:**

This paper presents early results towards a decentralized actor critic algorithm for fully observable, cooperative multi-agent tasks with theoretical guarantees. However, the reviewers raised many issues requiring significant further work which the authors acknowledged but could not complete due to the time limits of the ICLR review period. Hopefully this feedback is useful to the authors as they continue to develop this line of work for a future publication.